# Time-Restricted G-Protein Signaling Pathways via GPR176, G_z_, and RGS16 Set the Pace of the Master Circadian Clock in the Suprachiasmatic Nucleus

**DOI:** 10.3390/ijms21145055

**Published:** 2020-07-17

**Authors:** Shumpei Nakagawa, Khanh Tien Nguyen Pham, Xinyan Shao, Masao Doi

**Affiliations:** Department of Systems Biology, Graduate School of Pharmaceutical Sciences, Kyoto University, Sakyō-ku, Kyoto 606-8501, Japan; nakagawa.shumpei.64u@st.kyoto-u.ac.jp (S.N.); nguyen.tien.33r@st.kyoto-u.ac.jp (K.T.N.P.); shao.xinyan.23a@st.kyoto-u.ac.jp (X.S.)

**Keywords:** circadian clock, orphan GPCR, suprachiasmatic nucleus, GPR176, G-protein, RGS16, G_z_, *N*-glycosylation

## Abstract

G-protein-coupled receptors (GPCRs) are an important source of drug targets with diverse therapeutic applications. However, there are still more than one hundred orphan GPCRs, whose ligands and functions remain unidentified. The suprachiasmatic nucleus (SCN) is the central circadian clock of the brain, directing daily rhythms in activity–rest behavior and physiology. Malfunction of the circadian clock has been linked to a wide variety of diseases, including sleep–wake disorders, obesity, diabetes, cancer, and hypertension, making the circadian clock an intriguing target for drug development. The orphan receptor GPR176 is an SCN-enriched orphan GPCR that sets the pace of the circadian clock. GPR176 undergoes asparagine (*N*)-linked glycosylation, a post-translational modification required for its proper cell-surface expression. Although its ligand remains unknown, this orphan receptor shows agonist-independent basal activity. GPR176 couples to the unique G-protein subclass G_z_ (or G_x_) and participates in reducing cAMP production during the night. The regulator of G-protein signaling 16 (RGS16) is equally important for the regulation of circadian cAMP synthesis in the SCN. Genome-wide association studies, employing questionnaire-based evaluations of individual chronotypes, revealed loci near clock genes and in the regions containing *RGS16* and *ALG10B*, a gene encoding an enzyme involved in protein *N*-glycosylation. Therefore, increasing evidence suggests that *N*-glycosylation of GPR176 and its downstream G-protein signal regulation may be involved in pathways characterizing human chronotypes. This review argues for the potential impact of focusing on GPCR signaling in the SCN for the purpose of fine-tuning the entire body clock.

## 1. Introduction

G-protein-coupled receptors (GPCRs) constitute the largest family of cell surface receptors that participate in signal transduction for a range of stimulants, including light, odorants, peptide and non-peptide neurotransmitters, hormones, growth factors, and lipids. Activated GPCR signals control various downstream processes, which include sensory transduction, cell–cell communication, neuronal transmission, and hormonal signaling. Consistent with their versatile roles in physiology and disease, GPCRs are the therapeutic targets of nearly a third of US Food and Drug Administration (FDA)-approved drugs [1,2]. Despite their prominent clinical relevance, of the 356 non-olfactory GPCRs encoded in the human genome, approximately 38% are still regarded as ‘orphans’ whose cognate ligands are not known [3]. Deciphering their cognate ligands and functions in health and disease is, therefore, crucial from the perspective of both basic research and drug discovery.

A set of principal circadian clock neurons that governs daily cycles in behavior and physiology resides in the suprachiasmatic nucleus (SCN), located in the anterior hypothalamus. By establishing the pace and phase of physiological rhythms in tune with the regular day/night cycles, the endogenous clock confers an adaptive advantage to the organism. The most prominent advantage of this clock is that it allows for proactive, rather than entirely reactive, homeostatic coordination of physiological functions. Considering these benefits, it is perhaps unsurprising that malfunctions of this endogenous timer, due to jet-lag, rotational shiftwork and the irregular late night and/or early morning lifestyles in our current society, result in the onset of various diseases and health problems. Indeed, one of the most important conceptual changes brought about by the analysis of circadian-clock-deficient mice is that abnormalities in the circadian clock are linked not only to sleep arousal disorders, but also to a broad range of common diseases such as high blood pressure, obesity, and cancer [4,5,6]. Drug efficacy and toxicity also change throughout the 24-h day in an endogenous clock-dependent manner [6]. This growing evidence supports the potential value of developing drugs that target the circadian clock.

GPR176 is an SCN-enriched orphan GPCR that sets the pace of the circadian clock [7]. GPR176 is *N*-glycosylated in vivo, and this modification is required for its proper expression and function [8]. GPR176 couples to G_z_, a specific subtype of G_i/o_ family members. Even in the absence of a known ligand, GPR176 possesses an agonist-independent constitutive activity that leads to reduced cAMP synthesis [7,9]. This review focuses on the function of GPR176 and its downstream signaling pathways regulating the circadian clock in the SCN. In addition to focusing on GPR176/G_z_, this review also emphasizes a role for the regulator of G-protein signaling 16 (RGS16) in the circadian regulation of G-protein-cAMP signaling in the SCN [10,11]. Interestingly, recent genome-wide association studies identified single-nucleotide polymorphisms (SNPs) in RGS16 and ALG10B, an enzyme involved in *N*-glycosylation, that are associated with human morningness/eveningness chronotype (the preference for early or late sleep timing) [12,13,14,15]. These associations suggest that pathways regulated by G-protein signaling through *N*-glycosylated GPR176 and RGS16 may be involved in the determination of human chronotype. Developing drugs that target the SCN remains an unfulfilled opportunity for circadian clock-based chronomedicine. In this framework, this review argues for the potential merit of focusing on GPCRs in the SCN with the purpose of remodeling the entire body clock.

## 2. The SCN Is the Principal Circadian Pacemaker in Mammals

A series of discoveries regarding the location and function of the SCN are reviewed extensively elsewhere [16]; hereafter, we summarize the key findings. First, a pioneer brain lesion study aimed at discovering the anatomical structure of the mammalian master clock led to the identification of a part of the brain called the SCN (Figure 1). As its name suggests, the SCN is located just above (“supra” to) the optic chiasm, the place where the optic nerves cross beneath the anterior hypothalamus. A complete bilateral lesion of the SCN results in arrhythmic behavior, but transplantation of fetal SCN tissue can restore circadian behavior [17]. Importantly, SCN transplants from *Tau* mutant hamsters, which display a short-period phenotype, restore rhythms in SCN-lesioned wild-type hamsters and confer a period length characteristic of the mutant donor [18], demonstrating that the period is encoded by the SCN. Furthermore, the implantation of wild-type SCN generates circadian locomotor activity rhythms in global-clock-gene-deficient mice, which have no other functional clock in the body [19]. Thus, the SCN alone is sufficient to drive organismal 24-h rhythmicity. Recent results from cell-type-specific conditional clock-gene knockout mice and tissue-specific rescue or reconstitution of the clock gene in otherwise arrhythmic mice further support the assertion that the SCN is the center of the circadian clock [20,21,22,23,24,25]. 

At the molecular level, the rhythm-generating mechanisms of the circadian clock involve a set of core clock genes, which regulate their own transcription in a negative transcription–translation feedback loop [26]. Notwithstanding the recent intriguing discoveries of circadian oscillations in peroxiredoxin superoxidation in transcriptionally incompetent red blood cells [27] and in the SCN [28], the consensus conjecture regarding the mammalian clockwork supports the transcription-based feedback model. Particularly, studies in mutant mice carrying a mutation only at the E′-box *cis*-element in the promoter of the mammalian core clock gene *Per2* [29] demonstrate that circadian *cis*-regulatory element-mediated *Per2* gene transcription is essential for sustaining molecular clock oscillations (see Figure 2) and inducing stable physiological rhythms in vivo. Similarly, deleting the *Cry1* intronic enhancer region, which contains a ROR-responsive element (RRE), shortens the period of locomotor activity rhythms [30]. Therefore, multiple loops that use different *cis*-elements appear to act in concert to generate circadian rhythmicity.

Beyond the cell-autonomous mechanism described above, circuit-level interactions in the SCN are also crucial to synchronize thousands of individual neurons into a robust and coherent daily timer [16,31]. SCN neurons connect with each other at least partly via cell–cell interactions through neuropeptide signals, including vasoactive intestinal polypeptide (VIP), arginine vasopressin (AVP), and gastrin-releasing peptide (GRP) [32]. Deficiency of either VIP or its cognate G-protein-coupled receptor, VIPR2, causes desynchronization among SCN neurons [33]. Neuropeptide/GPCR-mediated intercellular coupling mechanism is, therefore, a unique and necessary feature of the SCN neurons to create robust and coherent ensemble circadian oscillations.

## 3. GPR176 Is a G_z_-Linked Orphan GPCR that Sets the Pace of the SCN Clock

The SCN is the site of the circadian clock center, thus expressing all known genes that determine the circadian period of locomotor activity rhythms. Based on this assumption, we launched the SCN-GENE PROJECT [7], which focuses on genes that are highly expressed in the SCN. More specifically, to identify new GPCRs that tune the central clock, we undertook the “SCN-GPCR” screening [7], in which we searched for the GPCRs whose expression is enriched in the SCN, generated knockout mice of candidate GPCR genes, and investigated potential noticeable defects in their rhythms in behavior and physiology. Using this screening strategy, the orphan GPCR GPR176 and calcitonin receptor CALCR were identified to be relevant for circadian clock function in the SCN [7,34].

GPR176 is an SCN-enriched orphan GPCR that sets the pace of normal circadian behavior (see Figure 3) [7,35]. Genetic ablation of GPR176 in mice shortens the circadian period of locomotor activity rhythm. A short-period phenotype was also reproducibly observed for the rhythms in transcription of the core clock gene *Per1* in cultured SCN tissue slices obtained from the *GPR176* knockout mice. Thus, the defects in the SCN mirror the behavioral rhythm output.

Unlike GPR176, CALCR is not involved in circadian behavior [34]. Rather, CALCR is particularly important for the circadian body temperature rhythm during the night, which is the active phase for mice [34]. The roles of CALCR in temporal regulation of body temperature in mice and flies have been extensively reviewed elsewhere [36].

### 3.1. GPR176 Colocalizes with VIPR2 and Displays Oscillatory Abundance

GPR176 is an evolutionarily conserved, vertebrate class-A orphan GPCR (GPCRdb, https://www.gpcrdb.org/). In the mouse SCN, nearly all neurons express GPR176, which colocalizes with the VIP receptor VIPR2 [7].

Temporal expression profiles of GPR176 and VIPR2 reveal that, although these two GPCRs are expressed in the same SCN neurons, their peak expression phases are opposite: GPR176 expression peaks at night [7], whereas VIPR2 expression peaks during the day [37]. The presence of a conserved RRE in the promoter region of *GPR176* is consistent with its mRNA and protein peak expression during the night [7].

### 3.2. Agonist-Independent Activity of GPR176 Counteracts VIPR2-cAMP Signaling

Almost all GPCRs display two distinct structural conformations: an active and an inactive form. Agonists stabilize the receptor in its active conformation, whereas inverse agonists stabilize it in its inactive formation. Even in the absence of ligands, GPR176 spontaneously switches between these two states, generating agonist-independent basal activity [7]. Comparative studies among the class-A GPCRs [9] revealed that GPR176 is one of the most potent GPCRs in terms of agonist-independent basal activity.

The basal activity of GPR176 attenuates cAMP production (see Figure 4). This cAMP-inhibitory pathway therefore likely constitutes a “negative” limb that could counteract “positive” cAMP signaling via the VIP-VIPR2 axis in the SCN. Consistent with this “yin-yang” model (see Figure 3), the genetic deletion of GPR176 in the SCN reduces the suppression of cAMP signaling during the night [7].

Circadian fluctuation of cAMP signaling has been shown to be crucial for maintaining circadian clock function in the SCN [38]. The possible downstream pathway(s) of cAMP concentration rhythms may involve transcriptional regulation of *Per1* through a cAMP/Ca^2+^-responsive element on its promoter. However, the molecular mechanisms or circuits through which the circadian cAMP signal is integrated into the hardwired SCN clock are still not fully understood [31].

### 3.3. The Unique G-Protein Subclass G_z_ Is Required for GPR176 Basal Activity

The unique G-protein subclass G_z_, also known as G_x_, has been shown to be a specific G-protein partner for GPR176 [7]. G_z_ is an evolutionarily conserved G-protein subtype that belongs to the G_i/o_ family. Similar to G_i1_, G_i2_, and G_i3_, G_z_ inhibits the activity of adenylyl cyclases and thereby reduces cAMP synthesis. However, G_z_ is not just an “ersatz” G_i_. As its name suggests, G_z_ displays a range of unique properties that set it apart from other G_i/o_ members [39]. First, in contrast to the ubiquitous expression of G_i_, tissue distribution of G_z_ is restricted to endocrine tissues and the brain, including the SCN [7,39]. Second, unlike G_i_, pertussis toxin (PTX) does not inhibit G_z_ [39]. PTX catalyzes ADP ribosylation at the conserved cysteine in the fourth position from the C terminus of G_i/o_ proteins. The absence of this cysteine residue renders G_z_ insensitive to this drug, which is congruent with the resistance of GPR176-G_z_ activity to PTX (see Figure 4B,C). Finally, as compared to other G_i_ members and G_s_, G_z_ bears relatively low intrinsic GTPase activity [40,41,42]. The *k*_cat_ values for GTP hydrolysis by G_s_ and G_i/o_ are in the range of 1–5 min^−1^ (Table 1). By contrast, G_z_ shows a rate of activity that is approximately 100 times slower (0.05 min^−1^). Thus, once GTP binds to it, the activated G_z_ does not readily return to the inactive state, which results in a long-lasting signal. This characteristic suggests that G_z_ may be suitable for hourly control of the circadian clock.

In addition, the gene encoding G_z_ (Gnaz) shows rhythmic expression in the retina and pineal gland [43]. Neither the visual α-transducin (i.e., Gnat and Gnat2) nor any other G-protein α subunits (Gnas, Gnal, Gnai1, Gnai2, Gnai3, Gnaq, Gna11, Gna12, Gna13, Gna14, Gna15) display circadian expression, further suggesting that G_z_ is specifically involved in circadian clock-related regulation. G_z_ can also partner with other GPCRs than GPR176. Among the other GPCRs that are identified to be expressed in the SCN, the serotonin 5-HT1A receptor [44,45] and melatonin MT1 and MT2 receptors [46,47] can also couple to G_z_, which extend the potential routes of contribution of G_z_ to the circadian clock mechanism.

### 3.4. GPR176 Is an N-Glycosylated GPCR

In the SCN, GPR176 undergoes *N*-glycosylation [8]. *N*-glycosylation is one of the most common posttranslational modifications that affects GPCR and other membrane protein function. However, the literature regarding the presence and functional role(s) of *N*-glycosylation of orphan GPCRs is still sparse (see references in [8,48,49]). The orphan receptor GPR61 has been recently reported to be *N*-glycosylated, but this modification is not vital for its expression [49]. By contrast, *N*-glycosylation of GPR176 is required for the efficient expression of this protein [8] (Figure 4D). 

*N*-glycosylation occurs at four conserved asparagine residues in the N-terminal extracellular region of GPR176 (see Figure 4D). The prevention of *N*-glycosylation by mutation of these sites results in a drastic reduction in GPR176 protein expression (Figure 4D). Normally, GPR176 functions as a cell-surface membrane protein. However, non-glycosylated mutant proteins are mostly retained in endoplasmic reticulum (ER) and degraded in part through a proteasomal pathway. At the molecular level, deficient *N*-glycosylation does not impair the constitutive activity of GPR176 [8]. However, attenuated protein expression of GPR176 due to the lack of *N*-glycosylation leads to reduced total cAMP-repressive activity in the cells (see Figure 4E).

To date, over 330 non-synonymous SNPs in the *GPR176* gene have been identified (dbSNP, https://www.ncbi.nlm.nih.gov/), but only a few have been characterized biochemically or structurally. Importantly, missense variations in the conserved *N*-glycosylation sites of human GPR176 (SNPs rs1473415441 and rs761894953) affect *N*-glycosylation and thereby reduce protein expression and cAMP-repressive activity [8]. However, it is important to note that, although functional, these two SNPs occur only very rarely (minor allele frequency < 0.001%) and have not been reported to correlate with disease. The (patho)physiological contribution of *N*-glycosylation of GPR176 is thus unclear at present. Generally, owing to the problem of statistical power, rare-variant effects cannot easily be identified via SNP association studies.

Recent genome-wide association studies of approximately 100,000 individuals revealed the association of a common SNP for *ALG10B* (minor allele frequency, 48%) with human circadian behavior [12], raising interest in understanding the impact of *N*-glycosylation on the circadian clock. While the mechanism(s) by which ALG10B affects the circadian clock have yet to be clarified, it is tempting to assume that this enzyme affects the extent of *N*-glycosylation of GPR176. In addition, other clock-related GPCRs and non-GPCR proteins that operate in the SCN might be similarly *N*-glycosylated in vivo (see Figure 5). Intriguingly, in vitro studies suggest that VIPR2 [50], vasopressin receptor V1A [51] (which is involved in phase recovery during jet lag [52]), as well as the ion channels NMDA receptor [53] and AMPA receptor [54] (both contributing to light-induced glutamatergic phase-resetting of the circadian clock [16]) are also likely to be *N*-glycosylated. How ALG10B-mediated *N*-glycosylation affects the physiological function of GPCRs and non-GPCRs in the SCN will be the topic of future studies (Figure 5).

## 4. RGS16 Is a Regulator of G-Protein-cAMP Signaling in the SCN

RGS proteins constitute a family of over 20 members that participate in the regulation of G-protein signaling. The enrichment of RGS16 mRNA and protein expression in the SCN [10] prompted researchers to test for the role of this specific subtype RGS in the regulation of the circadian clock. Knockout and knockdown of RGS16 both compromise the periodicity of behavioral rhythms [10,11]. Loss of RGS16 leads to impaired circadian generation of intracellular cAMP in the SCN [10], indicating an intimate connection between RGS16 and GPR176/VIPR2-mediated cAMP signaling (see the model in Figure 3). 

Biochemical in vitro studies show that RGS16 is a GTPase-accelerating protein for G_i_ [55]. By promoting GTP hydrolysis, RGS16 terminates GTP-associated G_i_ signaling; therefore, this protein acts as a negative modulator for G_i_ signaling. The most interesting feature of RGS16, which distinguishes it from other RGSs, is that it functions in the SCN in a time-of-day-specific manner [10,11]. A direct transcriptional control by the circadian clock components through D-box and E-box allows the mRNA and protein expression of RGS16 to simultaneously change over time, with a peak in the morning. Systematically, this may help to gate SCN cAMP signaling in a timely manner, with the gate of cAMP signaling open at the beginning of the day and closed during the night (Figure 3). RGS16 thus likely acts as a rheostat that gates cAMP signaling in the SCN. A question remains open as to whether RGS16 dictates G_z_ signaling in the SCN.

The potential importance of RGS16 in human biological clock function has been discovered by three independent research teams, including 23andMe Inc., a company that offers personal genomic DNA testing in the US. The teams interrogated human genomic information from about 100,000 individuals whom they queried about their propensity to get up early or sleep late in the morning. All three teams, intriguingly, reached the same conclusion that genetic variants of *RGS16* are closely associated with waking up early [12,13,14]. Moreover, a follow-up association study on more than 650,000 individuals further confirmed this tight linkage between *RGS16* and human chronotypes [15]. Although more studies are required, current evidence suggests that the RGS16-dependent circadian mechanisms that were originally reported in rodents likely also operate in humans. 

## 5. Advantage of Focusing on GPCRs in the SCN

To enhance discussion on a future direction, in the following section, we elaborate on what we consider as the main advantage of focusing on GPCRs in the SCN. 

We chose this tissue-specific approach with the longer-term goal of identifying novel targets for circadian chronotherapy [7]. Generally, the discovery of the key “common” clock components that constitute the circadian clock has advanced through different approaches, including mutagenesis-based forward genetics with model organisms, biochemical approaches, followed by confirmatory reverse genetics, in silico studies, and cell reporter-based screening (see references in [26]). In contrast to these approaches, however, not all genes identified by the SCN-GENE PROJECT may be important for the circadian clock function outside the SCN. In the longer term, this would provide a beneficial option for therapeutic development. The clocks outside the SCN are not functionally uniform and are heterogeneously distributed across the body. Studies in primates [56] and rodents [57] demonstrate that the phase of expression of circadian clock components in the SCN differs from that in peripheral tissues. Therefore, it is conceivable that any drug that targets the common clock components would reset all the clocks simultaneously and equally, which may be deleterious for maintaining the adaptive phasic order between SCN and peripheral tissues (unless such drugs were to be delivered to the target tissue selectively, which would be a very demanding task). On the other hand, SCN-specific components, such as GPR176, would enable the design of compounds that affect the SCN while having no direct action on peripheral clocks, leaving them subject to SCN-dependent cues, and thereby retaining relative circadian phases and amplitudes. Therefore, we reason that, in combination with the other methods, the research focusing on GPCRs and other drug targets in the SCN will provide an alternative, valuable route to develop drugs that harmonize the whole body clock.

## 6. Deorphanizing GPR176

High-throughput ligand searches for orphan GPCRs have been performed using various methods, such as PathHunter and PRESTO-Tango β-arrestin recruitment assays, as well as time-resolved FRET-based internalization assay [58,59,60,61]. Nevertheless, GPR176 ligands have not been identified to date [58,59,60,61]. Based on this, one possibility is that there may be no endogenous ligand for the orphan receptor GPR176. In this case, its activity would mainly be regulated via the levels of receptor protein expression, which, for GPR176, is highly circadian in the SCN. Consistent with its negative effect on cAMP, GPR176 protein expression increases at night, a phase when cAMP levels in the SCN are decreased to the circadian nadir levels [10,38]. In this scenario, the agonist-independent spontaneous activity of GPR176 could explain its physiological function in vivo. 

## 7. Perspectives

GPR176 is an orphan receptor enriched in the SCN, where it controls the pace of the circadian clock. We reviewed, here, the most recent evidence that supports a pivotal role of GPR176 and its associated partners and downstream pathways in the regulation of the endogenous circadian clock. GPR176 is currently still an orphan with no ligand. The presence of unidentified endogenous ligands is always difficult to exclude, particularly for receptors with constitutive action. However, even in the absence of a known endogenous ligand, small molecules acting as surrogate ligands could still be developed for orphan GPCRs, which highlights their suitability for drug discovery. A key impediment to ligand screening for orphan GPCRs is the uncertainty about the downstream G-protein partner via which they signal. In this regard, the stable cell clones that we developed for assaying GPR176-G_z_ signaling [8] (Figure 4) may be useful for activity-based ligand screening for GPR176. Finally, site-specific expression is another important feature of GPR176. In an optimistic view, a drug that specifically targets the SCN may help normalize circadian sleep disorders and its associated diseases, with fewer potential side effects on peripheral clock functions.

## Figures and Tables

**Figure 1 ijms-21-05055-f001:**
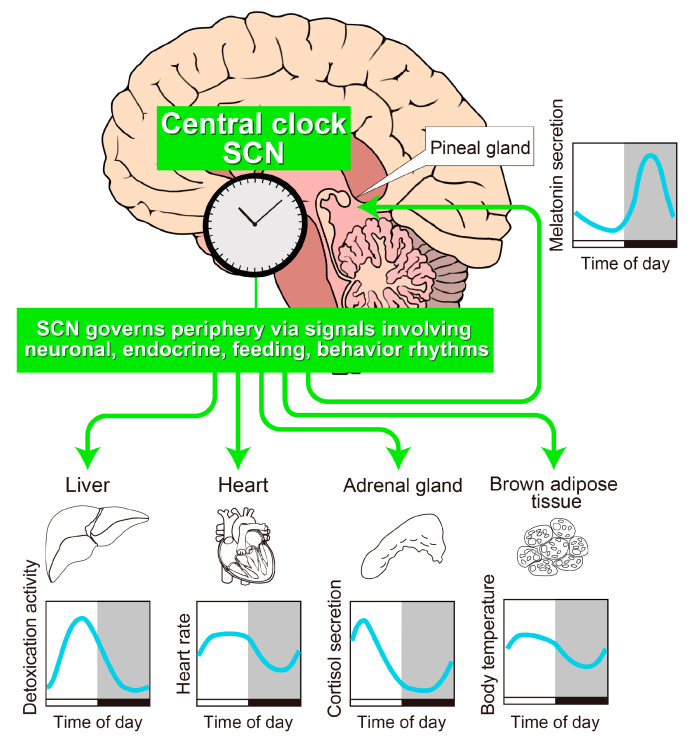
A schematic drawing of the architecture of the body clock. Most physiological processes exhibit circadian oscillations, which are synchronized by a central pacemaker located in the suprachiasmatic nucleus (SCN) of the anterior hypothalamus. Physiological rhythms in peripheral tissues such as liver, heart, adrenal gland, brown adipose tissue, and pineal gland are synchronized by an array of direct or indirect signals from the SCN. White and black bars indicate 12-h light/12-h dark cycle. Gray shading indicates the dark phase.

**Figure 2 ijms-21-05055-f002:**
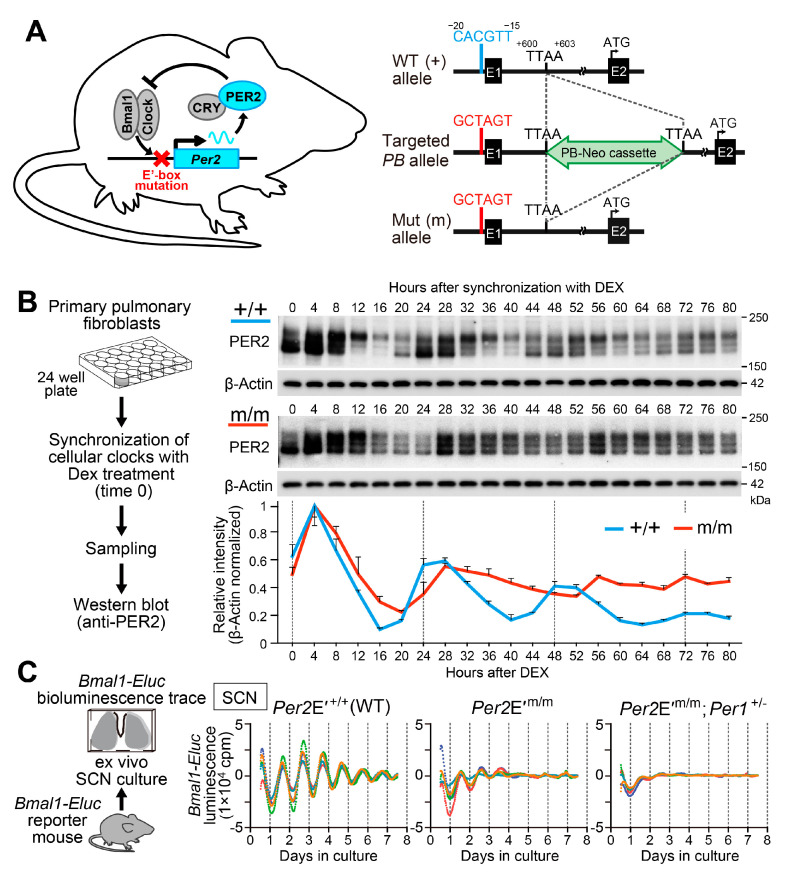
Targeted disruption of *Per2* E′-box leads to unsustainable gene expression rhythms in organotypic SCN slices and cultured lung fibroblasts. (**A**) Generation of *Per2* E′-box mutant mice. The CACGTT *Per2* E′-box was mutated to GCTAGT using the *piggyBac* transposon system. Numbering shows the position relative to the putative transcription start site (+1). E1, exon 1; E2, exon 2; Green, *piggyBac* transposon carrying a neomycin-resistant gene (PB-Neo cassette) inserted at a genomic TTAA site (+600 to +603). (**B**) Temporal profiles of PER2 protein expression in *Per2*E′^+/+^ and *Per2*E′^m/m^ lung fibroblasts. Representative immunoblots and normalized densitometry values (*n* = 3, mean ± s.e.m.) are shown. (**C**) *Bmal1-Eluc* bioluminescence traces of ex-vivo SCN cultures from *Per2*E′^+/+^ (*n* = 5), *Per2*E′^m/m^ (*n* = 5), and *Per2*E′^m/m^;*Per1*^+/−^ (*n* = 5) mice. Averaged de-trended data are shown. All tested SCN slices from *Per2* E′-box mutant mice displayed attenuated luminescence rhythms, which were damped within 2–3 cycles. The residual damping rhythms were dependent on *Per1*, because the luminescence from SCN slices of *Per2*E′^m/m^; *Per1*^+/−^ mice exhibited even fewer persistent rhythms. Data in (**B**) and (**C**) are reproduced from [29].

**Figure 3 ijms-21-05055-f003:**
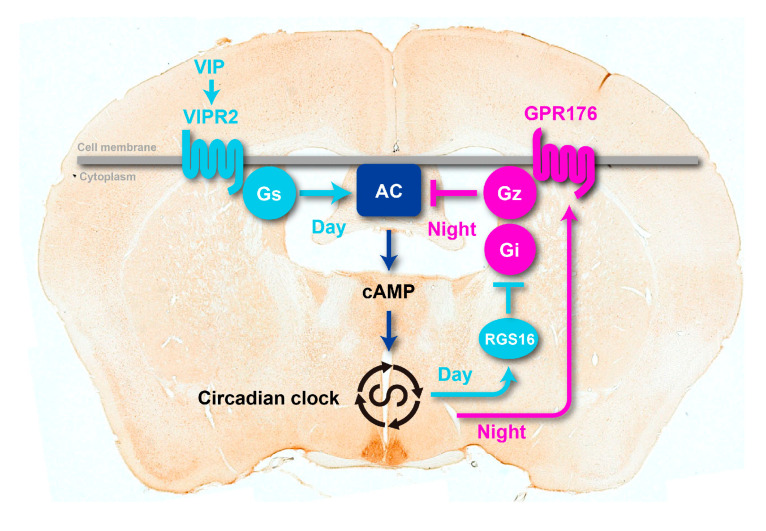
A model of circadian GPCR-cAMP signaling in the SCN. The orphan receptor GPR176 couples to G_z_ and thereby antagonizes the VIP-VIPR2-G_s_-mediated cAMP signaling that controls the intracellular circadian clock. The clock enhances expression of GPR176 at night and represses it during the day. At the beginning of the day, the clock induces expression of RGS16, which deactivates the cAMP-repressive signal through G_i_ and perhaps G_z_. This helps increasing cAMP synthesis at the start of the day. AC, adenylate cyclase. The photograph behind the schematic model is a representative mouse coronal brain section immunolabeled for GPR176.

**Figure 4 ijms-21-05055-f004:**
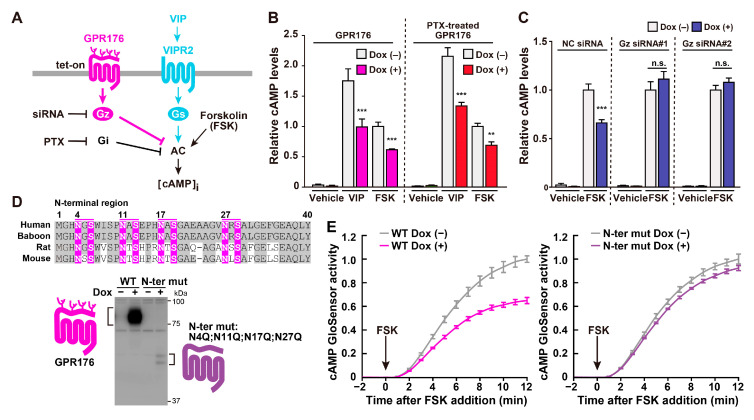
Biochemical features of GPR176: agonist-independent constitutive activity, G_z_ coupling, and *N*-glycosylation. (**A**) Generation of GPR176 tet-on cells constitutively expressing VIPR2. (**B**) Antagonistic activity of GPR176 on VIP- and forskolin (FSK)-stimulated cAMP accumulation in GPR176 tet-on cells. Cells of the same batch were cultured in parallel with or without Dox for 24 h, then resuspended in assay buffer for 1 h and stimulated with VIP or FSK for 15 min. Where specified, cells were treated with PTX. cAMP levels were determined by enzyme immunoassays. (**C**) Interference of GPR176 activity by G_z_ knockdown. Cells were transfected with two different siRNA mixtures targeting G_z_. NC, negative-control siRNA. (**D**) The potential *N*-glycosylation sites in the N-terminal region of human, baboon, rat, and mouse GPR176. Sequences that match the consensus *N*-linked glycosylation motif (N-X-S/T, where X ≠ P) are highlighted in magenta. Immunoblots of human GPR176 and its mutant, which lacks the N-terminal potential *N*-glycosylation sites (N-ter mut: N4Q;N11Q;N17Q;N27Q) are shown below the alignment. Note that the mutation reduces the size and abundance of GPR176. (**E**) FSK-induced cAMP GloSensor luciferase activity traces in Dox-treated (+) and untreated (−) WT (*left*) and N-ter mut (*right*) GPR176 cells. The arrow indicates the start of FSK treatment. Notably, attenuated N-ter mut expression is accompanied by a reduced total activity of GPR176. The graphs shown in (**A**–**C**) and (**D**,**E**) are modified from [7] and [8], respectively.

**Figure 5 ijms-21-05055-f005:**
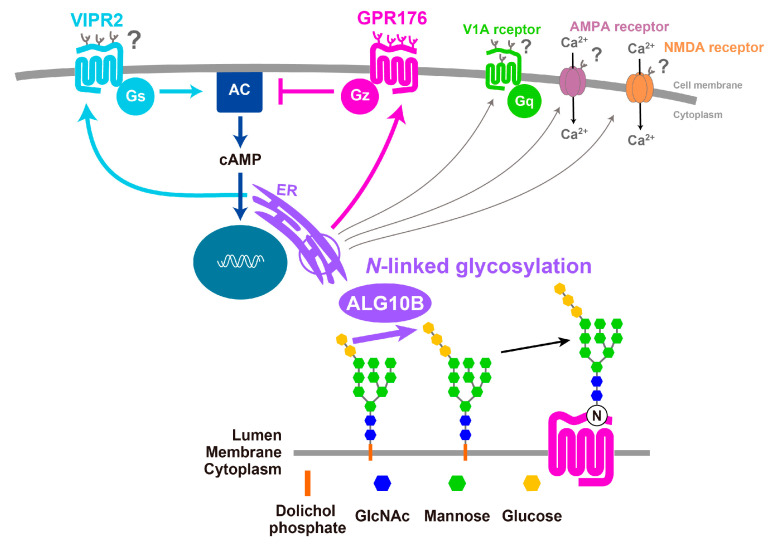
Possible roles of ALG10B-mediated *N*-glycan modification in the SCN. ALG10B is an enzyme catalyzing the addition of the terminal glucose residue to the growing *N*-glycan in the endoplasmic reticulum (ER). *N*-glycosylation occurs not only on GPR176 but also on other clock-related GPCRs and non-GPCR membrane proteins.

**Table 1 ijms-21-05055-t001:** *k*_cat_ for GTP hydrolysis by Gα subtypes.

Subtype	*k*_cat_ (min^–1^)	Source	Ref
Gα_s_	4.5	bovine, recombinant, purified from *E*. *coli*	[40]
Gα_i1_	2.4	rat, recombinant, purified from *E*. *coli*	[41]
Gα_i2_	2.7	rat, recombinant, purified from *E*. *coli*	[41]
Gα_i3_	1.8	rat, recombinant, purified from *E*. *coli*	[41]
Gα_o_	2.2	rat, recombinant, purified from *E*. *coli*	[41]
Gα_z_	0.05	human, recombinant, purified from *E*. *coli*	[42]

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
