# Peer review of "Time-Restricted G-Protein Signaling Pathways via GPR176, Gz, and RGS16 Set the Pace of the Master Circadian Clock in the Suprachiasmatic Nucleus"

_ijms, 2020, doi:10.3390/ijms21145055_

Round 1

Reviewer 1 Report

Nakagawa and colleagues provide a timely and comprehensive review on G-protein signaling in the SCN, the pacemaker of the circadian clock system. The paper is well-written and summarizes the state of knowledge, but also outlines recent research strategies and future directions. I have only minor comments:

  1. Line 37: use “disease” (singular)
  2. Lines 79 & 81: use “rhythms” (plural)
  3. Figure 1: the x-axis labels (0-24 referring to ZTs) are not very intuitive for non-chronobiologist readers. I would suggest removing the labelling and instead explain the black and white bars in the legend. Text in the lower green rectangle is somewhat hard to read (at least in my printout).
  4. Line 93: insert “the” before “rhythm-generating”
  5. Line 110: Would suggest the following wording: “Deficiency of either VIP or its cognate G-protein coupled receptor, VIPR2, causes…”
  6. Line 128: Would suggest the following wording: “The SCN is the site of the circadin center, thus, expressing all known genes…”
  7. Line 134: remove “/or”
  8. Line 135: remove “the” before “circadian”
  9. Line 140: would suggest writing “GPR176 knockout mice”
  10. Line 174: remove “the” after “maintaining”
  11. Line 188: there is an incomplete sentence because of the inserted figure which then continues as a new paragraph in line 206.
  12. Line 234: insert “the” before “GPR176
  13. Line 251: would suggest removing “signal”
  14. Line 252: remove “the” before “GPCRs”
  15. Line 272: Would suggest the following wording: “… to simultaneously change over time with a peak in the morning.”
  16. Line 275: remove “the” before “cAMP”
  17. Line 277: insert “function” after “biological clock”
  18. Line 307: should read “Deorphanizing”

Reviewer 2 Report

The manuscript is well-written and fits the scope of the journal. The authors present a nice overview of GPR76 and its signal transduction pathway and outline evidence supporting the importance of this receptor as a novel therapeutic target for circadian disorders. The manuscript was well organized, however, there are grammatical errors and confusing wording that make some sections harder to understand. I recommend a careful review the manuscript to correct these errors. I have noted some, but not all, of these errors below.

  • Line 128 is not clear, what is a “circadian center”
  • line 13 punctuation
  • Line 15- sentence “sleep-wake disorders” – add an s to disorders
  • Line 20 Gz needs to be defined in abstract.
  • Line 22 Commas around “employing … individual chronotypes”
  • Line 71 and line 127 are the same. The sections have the same title.
  • Line 91 should be “adrenal gland”
  • Line 93 change mechanism to mechanisms
  • Line 185 change brain to “the brain”
  • Line 206 other should be lower cased and indention should be removed to continue paragraph from line 188.

Reviewer 3 Report

The review by Nakagawa et al., is well written, and their article has comprehensively reviewed how GPR176 and correlated signalling contributes to circadian rhythms in the SCN master clock. The sections are well organized: from a general introduction on GPCR and the importance of the SCN clock, to specific information on GPR176 regulation and signalling pathway.

I believe this review will give a great contribute to the scientific literature in the area of the role of molecular mechanisms underlying circadian rhythms.
